# UBE3A: The Role in Autism Spectrum Disorders (ASDs) and a Potential Candidate for Biomarker Studies and Designing Therapeutic Strategies

**DOI:** 10.3390/diseases12010007

**Published:** 2023-12-27

**Authors:** Bidisha Roy, Enyonam Amemasor, Suhail Hussain, Kimberly Castro

**Affiliations:** Life Science Centre, Department of Biological Sciences, Rutgers University-Newark, Newark, NJ 07102, USA; ea538@scarletmail.rutgers.edu (E.A.); sah312@scarletmail.rutgers.edu (S.H.); kic20@scarletmail.rutgers.edu (K.C.)

**Keywords:** autism spectrum disorders, UBE3A, mechanisms, biomarker, therapy

## Abstract

Published reports from the CDC’s Autism and Development Disabilities Monitoring Networks have shown that an average of 1 in every 44 (2.3%) 8-year-old children were estimated to have ASD in 2018. Many of the ASDs exhibiting varying degrees of autism-like phenotypes have chromosomal anomalies in the Chr15q11–q13 region. Numerous potential candidate genes linked with ASD reside in this chromosomal segment. However, several clinical, in vivo, and in vitro studies selected one gene more frequently than others randomly and unbiasedly. This gene codes for UBE3A or Ubiquitin protein ligase E3A [also known as E6AP ubiquitin-protein ligase (E6AP)], an enzyme involved in the cellular degradation of proteins. This gene has been listed as one of the several genes with a high potential of causing ASD in the Autism Database. The gain of function mutations, triplication, or duplication in the UBE3A gene is also associated with ASDs like Angelman Syndrome (AS) and Dup15q Syndrome. The genetic imprinting of UBE3A in the brain and a preference for neuronal maternal-specific expression are the key features of various ASDs. Since the UBE3A gene is involved in two main important diseases associated with autism-like symptoms, there has been widespread research going on in understanding the link between this gene and autism. Additionally, since no universal methodology or mechanism exists for identifying UBE3A-mediated ASD, it continues to be challenging for neurobiologists, neuroscientists, and clinicians to design therapies or diagnostic tools. In this review, we focus on the structure and functional aspects of the UBE3A protein, discuss the primary relevance of the 15q11–q13 region in the cause of ASDs, and highlight the link between UBE3A and ASD. We try to broaden the knowledge of our readers by elaborating on the possible mechanisms underlying UBE3A-mediated ASDs, emphasizing the usage of UBE3A as a prospective biomarker in the preclinical diagnosis of ASDs and discuss the positive outcomes, advanced developments, and the hurdles in the field of therapeutic strategies against UBE3A-mediated ASDs. This review is novel as it lays a very detailed and comprehensive platform for one of the most important genes associated with diseases showing autistic-like symptoms. Additionally, this review also attempts to lay optimistic feedback on the possible steps for the diagnosis, prevention, and therapy of these UBE3A-mediated ASDs in the upcoming years.

## 1. Introduction

Published reports from the CDC’s **A**utism and **D**evelopment **D**isabilities **M**onitoring (ADDM) Networks predicted an average of 1 in every 44 (2.3%) 8-year-old children to have an ASD in 2018. However, reports from the World Health Organization and worldwide estimates suggested 1 out of 100 children suffered from autism in the year 2020, and its occurrence shows a bias towards males, as the median male-to-female ratio was found to be 4.2 [1]. The ADDM Network data suggests that the prevalence of autism per 1000 children rose from 11.3 to 27.6 between 2000–2020 (https://www.cdc.gov/ncbddd/autism/data.html, 22 May 2023).

Many of the ASDs with reported autism-like phenotypes of varying degrees have been attributed to the involvement of chromosomal anomalies in the chromosome 15q11–q13 segment. Several putative candidate genes associated with ASD lie within this chromosomal segment. This suggests that epigenetic modifications in the chromosome 15q11–q13 region might be a contributing factor in the pathophysiology of ASD. However, several scientific analyses and research studies [Table 1 and Table 2, with supported references] using clinical, in vivo, and in vitro samples and varied techniques have shown that the UBE3A gene was selected in an unbiased and random manner more frequently than other genes. Our summary in Table 2 showed that out of 31 studies, which included clinical samples, in vivo, and in vitro models, 16 studies identified the UBE3A gene as one of the hits. This was followed by the GABRB3 gene with five hits, ATP10A with four hits, and ATP10C and MAGEL2 genes with two hits. The high occurrence of the UBE3A gene in various ASD-related studies prompted us to focus on this gene in this review. This review encompasses a detailed and comprehensive analysis of the UBE3A gene, its protein product, and its mechanistic role in the pathophysiology of ASD.

UBE3A or Ubiquitin protein ligase E3A is an enzyme involved in targeted protein degradation and maintains protein homeostasis within the cell. This gene has been listed as one of the numerous genes with a high potential of causing ASD in the Autism Database, [http://autism.mindspec.org/RescueModel/UBE3A/F_UBE3A_29_KO_HT_G-UBE3A, 22 May 2023; [33,34,35]]. Various mutations including the gain of function mutations, triplication, or duplication in the UBE3A gene are also associated with various ASDs like Angelman Syndrome (AS) and Duplication 15q Syndrome (Dup15Q Syndrome). These syndromes are identified by clinical symptoms like abnormalities in social interactions, intellectual disability, and various behavioral defects, like AS and Dup15q Syndrome [36,37,38,39,40,41,42,43]. The genetic imprinting of UBE3A in the brain and a preference for neuronal maternal-specific expression are the key features of various ASDs including AS and Dup15q Syndrome [44,45,46].

The Dup15q syndrome is a sporadic congenital disease affecting 1 in 30,000 to 1 in 60,000 children globally [47]. AS is also another uncommon neurodevelopmental disorder with an estimated prevalence of 1 in 12,000 to 1 in 20,000 people in the United States (National Organization for Rare Disorders, 2018, retrieved from https://rarediseases.org/rare-diseases/angelman-syndrome, 22 May 2023. Both of these diseases with autistic-like clinical manifestations have been extensively investigated and several genes have been linked and associated with them. However, a few genes like UBE3A and HERC2 were modulated in both these diseases at the transcriptional level [48].

UBE3A or Ubiquitin protein ligase E3A [also known as E6AP ubiquitin-protein ligase (E6AP)] is an enzyme involved in the cellular degradation of proteins. It is transcribed and translated from the UBE3A gene in humans. The degradation of proteins is a cell biological process and involves the removal of dysfunctional, worn out, abnormal, and unnecessary proteins and helps maintain cellular protein homeostasis and normal function. UBE3A labels all damaged or dysfunctional proteins with a small tag or marker protein called ubiquitin. These ubiquitin-tagged proteins are destined for a fate of degradation. Protein complexes such as proteasomes degrade dysfunctional proteins tagged with ubiquitin by proteolytic cleavage of peptide bonds [49,50,51].

Usually, the UBE3A gene actively functions in a large proportion of somatic cells in two copies. However, in neurons, only the copy inherited from the mother [referred to as the maternal copy] is functionally active. This phenomenon is also known as paternal imprinting [45,52,53]. On the other hand, some neurons and glial cells may exhibit a bi-allelic expression of UBE3A [54,55,56]. The silencing of the UBE3A gene on the paternal allele is thought to occur via a Large Non-Coding Antisense Transcript, known as the UBE3A-ATS part of a lincRNA called “LNCAT” [57].

The location of the UBE3A gene, on the long arm [also referred to as the q arm] of chromosome 15 between positions 11 and 13, is an important locus and has been widely studied for deciphering the link between the gene’s chromosomal location and the severity of ASDs [46]. Some neurodevelopmental disorders arise from genetic rearrangements (like duplications and deletions) occurring at the 15q11–q13 locus. These include various forms of ASDs like Prader–Willi syndrome (PWS), AS, and Dup15q Syndrome. The chromosome’s 15q11–q13 region contains several genes which are controlled by genomic imprinting, a genetic event in which genes are specifically expressed from one parental allele. Therefore, the genes that are subjected to modulation by genomic imprinting are functionally haploid and possess only a single functional copy of the respective gene [58]. The three previously mentioned neurodevelopmental disorders arise primarily from deletions or duplications that occur at the 15q11–q13 locus. PWS, AS, and Dup15q Syndrome are disorders that arise from the loss of function or over-expression of at least one imprinted gene. They each occur with a frequency of approximately 1/15,000 to 1/30,000 live births globally [58,59].

PWS is caused by the loss of the paternally inherited chromosome 15q11.2–q13. The loss of expression from this chromosomal region generally occurs by one of the following three mechanisms: (1) 70% of individuals with PWS have a deletion of the entire genetically imprinted 15q11–q13 region, (2) 25% of individuals with PWS have maternal uniparental disomy (matUPD), in which both copies of chromosome 15 have been inherited from the mother, and (3) around 5% of individuals with PWS have an imprinting defect that causes paternal 15q11–q13 to function as though it were inherited from the mother [60]. PWS is characterized by hypotonia, thriving failure, feeding difficulties, hypogonadism, obesity, and hyperphagia [58]. There is no example of a point mutation in any gene causing PWS, suggesting that PWS is a contiguous gene syndrome, caused due to the loss of more than one gene.

AS is caused by the deficiency of functional maternal UBE3A [61,62]. AS is triggered due to one of four mechanisms: (1) the deletion of maternal 15q11.2–q13 in 74% of individuals, (2) the loss of the function mutation of maternal UBE3A in 11% of individuals, (3) paternal uniparental disomy (UPD), found in 8% of individuals, and (4) the imprinting defect that is found in 7% of individuals with AS [63]. The presence of individuals with AS due to point mutations in the maternal copy of the UBE3A gene validates that AS is a single-gene disorder, although other genes in the deletion region may play a partner-in-crime role in enhancing the severity of the disease. AS is characterized by developmental delay, ataxia, tremors, speech anomalies, a happy demeanor exhibited by hand flapping, and frequent smiling or laughter [58].

Dup15q syndrome exists in two forms: commonly occurring iso-dicentric chromosome 15q [idic15] and a rarer maternal interstitial duplication [intdup15] [41,64]. Individuals with idic15 have a small supernumerary chromosome holding two extra copies of the maternal 15q11.2–q13 region in a tail-to-tail orientation in addition to the two copies of normal chromosome 15. The idic15 chromosome also contains two centromeres, but is stable, owing to the dormancy of one of the centromeres. Individuals with idic15 are tetrasomic for the 15q11.2–q13 region, having three maternal copies and one paternal copy of the locus. On the other hand, individuals with the maternal intdup15 consist of one extra copy of the maternal 15q11.2–q13 region inserted in an inverted form and in tandem with another copy of the region. Thus, these individuals are triploid for the 15q11.2–q13 region and comprise two maternal copies and one paternal copy of the locus. Individuals with paternal interstitial duplications exist in the population with a milder phenotype compared to individuals with maternal interstitial duplications. Dup15q syndrome is characterized by hypotonia, developmental delay, intellectual disability, epilepsy, and autism [58].

Genetic mutations within the UBE3A locus are accountable for some cases of AS and PWS. These genetic mutations result in an unusually stunted, dysfunctional, and deformed UBE3A. Due to the inactivity of the paternal copy of the UBE3A gene in the brain, a genetic alteration in the residual maternal copy leads to the formation of defective or inactive enzymes in the brain leading to the various clinical conditions in ASDs. The UBE3A’s genetic location within the human chromosomal region 15q11–q13 is linked to various abnormalities in this region, like deletions and rearrangements (translocations) of genetic material. This has the potential to cause Angelman Syndrome. These chromosomal alterations prevent functionally active UBE3A from being produced in the brain. The 15q11.2–q13.1 duplication syndrome (dup15q syndrome) results from duplications of a specific portion of the 15q11.2–q13.1 chromosome, also referred to as the Prader–Willi/Angelman critical region (PWACR), and these duplications exist in two forms [65]. However, in any individual with the dup15q syndrome, only one of the two forms, either the idic15 or the intdup15, exists to cause the clinical symptoms.

In this review, we have attempted to summarize the structure and function of the UBE3A gene and its protein product; UBE3A’s mechanistic role in the various autism spectrum disorders (ASDs); the role of the 15q11–q13 region and the various genetic perturbations occurring within this chromosome 15 locus causing the ASDs; the potential usage of UBE3A in ASD biomarker studies; and the existing therapeutic interventions against UBE3A-mediated neurodevelopmental disorders. We hope that our review will provide insightful knowledge and scientific information regarding the relevance of UBE3A in ASDs, and pave the way for designing future effective and potent diagnostics and therapeutic strategies.

## 2. UBE3A—Structure and Function

An individual’s brain contains billions of neurons connected via trillions of synapses. This requires proper connection and development of the neurons for one’s brain to function normally. The disruption of these developmental processes leads to the development of neuron abnormalities in neurodevelopmental disorders like ASDs [66]. The genetic basis of ASDs is highly diverse because many heterogeneous genes are involved in causing the disease. However, many of these genes express themselves at the initial developmental stage of the disorder. A UBE3/E6-associated protein (E6AP) is one of the many genes with a predisposition to cause ASDs. The UBE3A, Ubiquitin Protein Ligase E3A, is one of the major genes implicated in autism spectrum disorder (ASD) [67] and encodes the E6AP protein, a protein that manifests itself in the brain in an imprinted manner. UBE3A is a gene, and E6AP is a UBE3A-protein product (Figure 1). This paper focuses on UBE3A as a potential biomarker by examining UBE3A’s neurobiological functions as a ligase of the Ubiquitin–Proteosome pathway. Several of E6AP’s target proteins with known functions are discussed, including SK2, p53, and Ephexin5 [68,69,70]. The UBE3A gene’s protein product is the E3 ligase or the E6-associated protein (E6AP), and genetic imprinting regulates its specific expression in the brain. Copy number variations (CNVs) leading to E6AP’s overexpression are strongly linked with ASD’s development, followed by clinical manifestations like reduced social communication, diminished social attentiveness, and increased monotonous behavior. Conversely, the loss of or reduction in E6AP’s expression leads to Angelman syndrome (AS), clinically described by speech anomalies, delayed motor development and function, and the occurrence of frequent, repetitive epileptic seizures [68,71,72,73].

UBE3A is a HECT domain E3 ubiquitin ligase responsible for protein ubiquitination, targeting them for proteomic degradation (Figure 1), [74]. It is generally associated with a neurodevelopmental disorder. E3 ligase, or E6-associated protein (E6AP), is the protein product of UBE3A, whose brain specificity is regulated through genomic imprinting as previously discussed. Differential splicing generates three potential E6AP proteins encoded by the UBE3A gene. The E6AP-coding region is 2.7 Kb long and contains ten exons or coding regions that encode 865 amino acids [75]. Neuronal activities change E6AP’s expression, particularly in cultured neurons, which is enhanced by either glutamate receptor activation or membrane depolarization. Various stimuli which are known to induce synaptic developmental processes in an experience-dependent manner also trigger E6AP’s expression.

The UBE3A gene instructs the synthesis of a protein known as ubiquitin-protein ligase E3A. These are enzymes that catalyze the degradation of other proteins within cells. Ubiquitin-protein ligases attach ubiquitin to proteins that require degradation. The ubiquitin-tagged proteins are then recognized and digested by cellular structures known as proteasomes (Figure 1). Proteolysis or protein degradation involves the removal of unnecessary and damaged proteins. This cellular process aids in maintaining protein homeostasis within cells [76,77]. Scientific research has shown that UBE3A plays an essential role in maintaining the normal function and development of the nervous system. Several studies suggest that UBE3A controls the balance of protein degradation and synthesis at the synapses [67]. Proteostasis regulation is crucial for synapses to adapt and change in response to experience and environment, a trait known as synaptic plasticity. Studies have found that synaptic plasticity is essential for memory and learning. Individuals inherit both copies of the UBE3A gene, with each parent contributing one copy each. These copies are activated in most of the body tissues. However, only the maternal copies are active in the spinal cord and the brain, with paternal copies lying in an active mode. This phenomenon is known as parent-specific gene activation and is also termed genomic imprinting. Normal brain development and functioning depend on proper UBE3A gene dosage, as shown by neurodevelopmental issues associated with mutation, deletion, and UBE3A copy number variations [36,37,78,79,80,81,82].

UBE3A gene dosage plays a role in ASD and AS [75]. The leading cause of Angelman Syndrome (AS) is mutations and deletions in UBE3A.

UBE3A’s functional loss in the brain leads to the development of Angelman Syndrome (AS), a sophisticated neuronal disease primarily affecting the neuronal system. UBE3A loses its function due to gene mutation or chromosomal changes that impact the maternal copy. Various genetic mechanisms can delete or turn off the UBE3A gene. In most cases involving Angelman syndrome a section of maternal chromosome 15 that contains the UBE3A gene is deleted. Other studies have also found Angelman Syndrome to result from UBE3A gene mutation [67]. UBE3A mutation results in the production of various versions of non-functional UBE3A. The copies inherited from the father are often inactive in some parts of the brain; therefore, when the maternal copy is deleted, it prevents the production of an enzyme that leads to the disease. The UBE3A gene’s gain-of-function mutation, triplication, and duplication are associated with autism, a condition clinically described by seizures, speech anomalies, and intellectual disability. Studies have found that UBE3A’s activity and expression should be controlled during the formation and growth of an individual’s brain. The reason behind this can be attributed to the pivotal role this gene plays in regulating synaptic plasticity and functions via proteasome-mediated degradation and targeted protein transcriptional regulation [75]. A genome sequencing study by Iossifov and colleagues [83] found that autism can be associated with a missense mutation [T485A] in the UBE3A gene. The T485A missense mutation affects the phosphorylation site for protein kinase A, known to impede UBE3A activities towards its substrates and itself. According to the International Classification of Disease, autism is a developmental disorder affecting the neuronal network with three important clinical features: repetitive, stereotyped behavior, impaired social skills, and interpersonal communication defects [75]. Most autism cases result from deleting maternally acquired copies of the 15q11–q13 chromosomal region. The 15q11–q13 chromosomal region consists of numerous genes, including the UBE3A gene (Figure 2). The ubiquitin–proteasome pathway, which is also known as the UPP, selectively regulates the levels and quality of different proteins by determining the quality and quantity of these proteins [84] and selectively removing damaged or unnecessary proteins to maintain normal cellular function. This pathway is important for the many functions of cells and the nervous system. During UPP processing and development, the protein substrate is identified by the bonding of a protein ubiquitin via the proteasome which breaks down the molecules of proteins. After initiating the transfer of the E2 enzymes, the E3 ligases which are 100 kDa proteins determine specific aspects of the ubiquitination. One of the major divisions of E3 ligases, HECT, plays an essential role in the regulation of sufficient amounts of UBE3A in brain processes. UBE3A is a gene that is also classified as a ubiquitin ligase [49]. Without proper levels of UBE3A by the HECT domain, an individual is likely to develop serious neurological disorders. Low and high levels of UBE3A or a lack of UBE3A function are linked to disorders such as Angelman syndrome (AS). Angelman syndrome is usually identified by intellectual disability, speech defects, and social and psychological deficits and can often present itself in hyperactive and abnormal patterns [85]. The UBE3A gene is located on the 15th chromosome of the human body [86]. The deletion of this chromosome often results in mental disabilities and developmental delays. An example of a disorder that is believed to be caused by an increased expression of UBE3A is ASD. UBE3A has also been linked to the control of Golgi acidification [87], levels of Cbln1 [88], and function of the mitochondria [89,90,91], and although the function and precision of UBE3A have been studied, the understanding of this brain function is still minimal. UBE3A is not only located on the 15th chromosome but it is specifically found on the 15q11–q13 part of the chromosome. The development of Angelman syndrome is not solely due to the mutation or deletion of UBE3A as many other genes are present spanning this chromosomal region. It was also identified in past studies that the mutation of the 15q11–q13 segment of the chromosome also affects the levels of autistic phenotypes that are displayed. For example, individuals who have an inverted duplicated 15q11–q13 segment usually show mild to moderate forms of autism whereas individuals with three copies of this part of the chromosome display more severe forms of autism spectrum disorder [36,37,78]. The initial finding regarding UBE3A was that this molecule was discovered to degrade the tumor suppressor P53 protein by interacting with the viral E6 oncoprotein in cells invaded by the human papillomavirus [74]. In addition to its ligase activity, UBE3A is known to act as a transcriptional co-activator [92,93,94], a cell cycle regulator [95,96,97], synaptic function, plasticity regulator [68,69,70,98,99,100,101], and cellular protein quality controller [102,103,104].

Alternative splicing of the UBE3A gene generates three E6AP protein isoforms. The coding region of the UBE3A gene is 2.7 Kb long, consisting of 10 exons and coding for a protein product comprising 865 amino acids. The second and the third isoforms have an extra 20 and 23 amino acids, respectively, at their amino terminuses. Isoforms 2/3 are similar in their E3 ligase catalytic site and activity. However, the correlation between the variable amino terminus and differential ubiquitination substrate specificity has not been elucidated and still needs further research. Recently, it was reported that UBE3A isoform 1 RNA is coded by a shortened sequence of the gene and lacks the sequence of the E3 catalytic domain. This UBE3A isoform 1 RNA is not translated into a protein within the detectable range. However, this RNA of UBE3A isoform 1 is an important regulator of spine maturation and complex dendrite formation. It was speculated that small non-coding RNAs, like micro-RNA 134 [105], might be a regulator of the UBE3A1 RNA, providing a unique mechanism for regulating the levels of UBE3A in the cells. Optimum levels of UBE3A are important for healthy brain development, as observed in neurodevelopmental disorders linked with various genetic aberrations like the deletions, mutations, and copy number variations (CNVs) of UBE3A.

A fly was also studied to understand UBE3A’s role in the glial function. Glia are known to act as accessory and supporting cells to the neurons in the nervous system [106]. The UBE3A gene in flies is known as the DUBE3A gene, and after the manipulation of the gene, the results observed in the fly are like that found in the animal model monitored. An overexpression of UBE3A (DUBE3A) also produces epileptic seizures, which is consistent with the overexpression or deletion of this gene in humans. In a study that mirrored the same method of increasing the expression of UBE3A in an animal, excessive amounts of UBE3A were found in the nucleus of mice, which affected the glutamatergic transmission negatively and resulted in an impairment in social skills, memory deficits, and other symptoms of Angelman syndrome [107,108]. These findings provide possible explanations of how an overabundance of UBE3A can be linked to autism spectrum disorder. However, it is important to acknowledge that although many of these findings in animal research are consistent with human responses to the manipulation of UBE3A, there are some differences in the expression of UBE3A between animals and humans. The functions of UBE3A extend beyond its link to autism spectrum disorder. For example, it has been observed that UBE3A also functions as a method of regulating synaptic function and synaptic plasticity, which encourages an expansion in memory and learning. UBE3A plays a role in these functions by interacting with the actions of the potassium channel and Arc protein by regulating its transcription process. The link between UBE3A and AS and other ASDs triggered a strong interest amongst researchers to understand the regulation of the synaptic function and plasticity of UBE3A. Numerous protein targets like cytoskeleton-associated protein (Arc), a Rho guanine nucleotide exchange factor (Ephexin5), and a small conductance calcium-activated potassium channel (SK2) [68,69,70] came to be noticed during this research investigation. It was found that UBE3A’s expression is enhanced during experience-dependent neuronal activity. The increased level of UBE3A thereby upregulates the formation of excitatory synapses by regulating the level of Arc, a synaptic protein that induces the internalization of AMPA types of glutamate receptor [68]. However, it was later found that Arc failed to act as a substrate of UBE3A; but, on the other hand, it was transcriptionally regulated by UBE3A [109]. UBE3A is known to tag SK2 with Ubiquitin and stimulate its endocytosis, resulting in increased NMDA receptor activation, thus modulating synaptic plasticity [70]. In the maternal UBE3A-deficient AS-modelled mice, many characteristic features of AS including cognitive and motor dysfunction, seizures, anxiety, aberrant circadian rhythm, and irregular sleep-wake cycles were exhibited [110,111,112,113]. Additionally, impairments in calcium/calmodulin-dependent protein kinase-II activity, long-term potentiation, experience-dependent synaptic plasticity, and imbalance of excitatory/inhibitory circuitry in the brain hippocampal regions [114,115,116,117] were also observed. This suggested that these abnormalities in AS mice could be due to aberrant levels of SK2, Ephexin5, Arc, or some other novel substrate of UBE3A.

## 3. The 15q11–q13 Region—The Regional Association of UBE3A in ASD

Multiple genes have been implicated in ASDs and incorporate a heterogeneous range concerning cellular functions. Most of the early developmental genes are involved in cell motility, cell adhesion, cytoskeleton formation, synapse development and maturation, and kinase activity. The UBE3A gene’s location on the front arm of the 15th chromosome at the q11–q13 site in humans [36,86] has been marked as an important site, showing UBE3A’s and several other genes’ linkage to ASDs. The loss of the maternally derived copy of the 15q11–q13 chromosomal region, owing to the deletion of this region, causes the majority of AS cases. Several genes encompass this region along with UBE3A [78] [Figure 2]. Many point mutations are involved in the UBE3A gene and account for 5–10% of cases of AS [36,37]. The brain-specific genomic imprinting of the UBE3A gene, within the chromosome region 15q11–q13, results in its maternal expression in the human fetal brain and adult cortex, and in the paternal copy being silenced [105].

Neurodevelopmental disorders occur due to epigenetic changes and genetic mutations in numerous imprinting groups. This special imprinting region in chromosome segment 15q11–q13 frequently overlaps in many of these disorders. For instance, the deletion of the 15q11–q13 segment (commonly known as the PWS critical region) in the paternal chromosome is associated with Prader−Willi syndrome (PWS) [118] and the deletion of the same segment in the maternal chromosome leads to Angelman syndrome (AS), as previously mentioned. Numerous disease symptoms in PWS and AS cases, irrespective of the involvement of the same segment, lie in the fact that PWS is caused by the deletion of specific paternally expressed genes like MAGEL2, MKRN3, PWRN1, NDN, C15orf 2, SNURF-SNRPN, and other snoRNA genes. On the other hand, AS is caused by the deletion of the maternally expressed UBE3A gene [119].

Maternal duplications of the 15q11.2–q12-imprinted region have been reported in certain cases of ASD [120,121]. Additionally, the interstitial duplication of the 15q11–q13 segment or an extra iso-dicentric 15 chromosomes [122,123] causes the 15q duplication syndrome. A comprehensive literature analysis across studies with the aim and objective of understanding the role of chromosome segment 15q11–q13 has indicated the involvement of a few specialized genes that can be considered potential candidates for ASD (see Figure 2). The list of these genes includes UBE3A and genes encoding GABA receptor subunits, such as GARBRB3, GABRA5, and GABRG3, etc.

A reasonably high percentage of genomic DNA copy number variations (CNVs) are present in 20% of cases of autism [79,80,81,82]. Previous reports have indicated that maternally acquired 15q11–q13 duplications and triplications are the most commonly occurring genomic CNVs in autistic patients [38,39,40]. Patients with an inverted duplication of an extra maternal 15q11–q13 segment (dup15) show mild to moderate autistic features, whereas patients with two extra copies (or triplication) resulting from an iso-dicentric extra-numerary chromosome (idic15) exhibit stronger symptoms. However, the paternal duplication of this region does not result in autism [41]. All these studies highlight the relevance of genetic imprinting within the duplicated chromosomal region as the causal factor of autism.

## 4. The Underlying Molecular Mechanisms of UBE3A-Mediated ASDs

ASDs include a spectrum of human neurological disorders with varied and diverse etiologies. The differential expression of UBE3A is one of the major causes of some of these ASD cases, but the mechanistic underpinning remains uncharacterized and elusive. Genetic transgenic models and ASD clinical cases with several wide ranges of transmuted genes revealed that the misregulation of signaling cascades, such as WNT, BMP, SHH, and retinoic acid signaling, might underlie the cause of ASDs. Additionally, during the period of autistic brain development, UBE3A has been known to impact and modulate WNT, BMP, and RA signaling pathways, indicating a cross-talk between UBE3A and these signaling pathways [124].

The variable expression of UBE3A is hypothesized to result in altered levels of the target proteins (Figure 3 and Figure 4, Table 3). Therefore, pinpointing the role of UBE3A targets in UBE3A-dependent pitfalls is necessary. Ephexin5 [a putative UBE3A substrate] with limited early developmental expression, modulates the formation of synaptic connections during the development of the hippocampus. Ephexin5 is aberrantly elevated in AS mice, artificially modeled by the genetic deletion of the maternally derived UBE3A gene. Sell et al. [125] have reported that Ephexin5 is a direct target of UBE3A-mediated ubiquitination, and lowering Ephexin5 levels in AS mice ameliorated several hippocampus-related behaviors, CA1 physiology, and alterations in the spine number of dendrites. These results proved Ephexin5 to be an important regulator of hippocampal deficits and other linked behavioral abnormalities in AS mouse models. Additionally, these results proved Ephexin5 and other UBE3A substrates to be targets for therapeutic intervention to ameliorate AS and other UBE3A-mediated developmental disorders. UBE3A’s dysregulation alters neuronal structure and synaptic functions, resulting in abnormal neuronal morphology, anomalous dendritic spine length, and neuronal polarization deformities with facts supported by scientific reports in AS transgenic model mice [44,126].

XIAP (X-linked inhibitor of apoptosis), known to regulate arborization patterns of dendrites, was identified as an E6AP target. The overexpression of neuronal UBE3A in ASD-model transgenic mice resulted in lower levels of XIAP in the neurons. This led to several other phenotypes like increased caspase activity, the breakdown of microtubules, defects in the maturation of dendritic spines, and reduced spine branching [127]. The AS transgenic mouse models also exhibited impaired learning and memory, synaptic plasticity, and long-term potentiation (LTP) defects in neurons of the hippocampal brain region [128], with additional features of reduced synaptic plasticity in the visual cortex [115,116]. Activity-Regulated Cytoskeleton-associated protein [ARC], a UBE3A target, internalizes α-amino-3-hydroxy-5-methyl-4-isoxazole propionic acid (AMPA) receptors. Therefore, increased ARC levels lead to a decrease in synaptic AMPA receptors in UBE3A-expressing AS neurons, resulting in synaptic dysfunction in transgenic AS models [68]. However, it is arguable whether ARC is a direct target of UBE3A, and this questionable hypothesis requires further investigation. There is an alternative theory of UBE3A negatively modulating ARCs transcriptionally [109]. The Small conductance Potassium Channel (SK2) is an important molecule known to induce LTP and aid in regulating plasticity in the neuronal synapse. The activation of the SK2 channel is dependent on N-methyl-D-aspartate (NMDA) receptor’s activation. UBE3A ubiquitinates SK2 and aids in its internalization. Elevated levels of SK2 have been observed in hippocampal neurons, with attenuated neuronal plasticity [70] in UBE3A-deficient AS transgenic model mice.

A previously discussed target of UBE3A, Ephexin5, is known to be a guanine nucleotide exchange factor (GEF) and is involved in the activation of RhoA. The proteolytic degradation of Ephexin5 aids in the formation of excitatory synapses. Elevated levels of Ephexin5 in AS neurons may lead to defects in synapse formation and cognition [69] and recurrent epileptic seizures [129]. An imbalance in excitatory and inhibitory synaptic transmission has been reported in AS-modelled mice which exhibit a higher predisposition to seizures [117,130]. In the UBE3A-deficient AS-modelled mice [131], inhibitory inputs from GABAergic neurons were reduced, leading to seizures. However, the addition of GABA agonists reduced the occurrence of seizure frequencies in these mouse models [132]. However, the UBE3A targets regulating GABAergic signaling have not been delineated and remain an unexplored arena for research in the future. Sleep abnormalities are reported in 75% of AS patients [133]. BMAL1 [Brain and Muscle ARNT-Like 1] is a protein vital for regulating the circadian clock system and is a UBE3A target [134]. Mice exhibiting UBE3A deficiencies demonstrate increased levels of BMAL1 with the phenotypic manifestation of abnormal defective circadian rhythms [113]. This possibly explains that sleep abnormalities reported in AS patients could be due to altered levels of the BMAL1 protein.

The Wnt signaling cascade is stimulated by UBE3A within the cell to set the homeostatic balance of β-catenin [135]. Wnt signaling regulates the general developmental processes of a cell and is implicated in ASD’s pathogenesis [136,137]. TSC2 regulates the mammalian target of rapamycin (mTOR) in a negative signaling cascade and has also been listed as one of the UBE3A targets [138]. Impaired mTOR signaling has been described in AS and ASD [139]. E3 ligase Ring1B [140], another UBE3A target, acts with the polycomb group repressor complex leading to the ubiquitination of histone H2A, thereby modulating global gene expression. A deficiency of UBE3A in mice led to high levels of Ring1B and H2A in Purkinje neurons present in the cerebellum of the brain, indicating their relevance in the manifestation of neuronal dysfunction, and other anomalies in AS. The transcription of the ESR2 gene [encoding ER-β] [141] is also regulated by UBE3A, and its presence elevates the levels of ER-Beta. ESR2 plays an important role in the development of the brain and is known to protect against neuronal degeneration. ESR2’s role in Alzheimer’s disease (AD) development and advancement has been well documented [142,143]. ESR2 regulates a brain-derived neurotrophic factor (BDNF), thereby regulating synaptic development, maturation, plasticity, and the formation of LTP [144,145]. Additionally, in a rat model of AD, the overexpression of ESR2 reduced the deposition of amyloid-β in the neuritic and amyloid beta plaques in the brain’s hippocampal region, leading to improvement in the learning, memory, and various other cognitive performances of these animals [142]. PSMD4 [26S subunit, non-ATPase 4] is one of the subunits of the 26S proteasome and has also been reported as a UBE3A target [146]. PSMD4’s ubiquitination and eventual degradation might affect the proteolysis of many vital proteins of the cells and affect global effects on the protein homeostasis in neurons. Recently, research showed that PSMD4 has a binding affinity to the AZUL domain of UBE3A, an important step for driving the UBE3A into the nucleus. This suggested that the nuclear localization of UBE3A is essential for neuronal development [147]. ALDH1A2, identified by Xu et al. and regulated by UBE3A negatively, is one of the important enzymes involved in the synthesis of retinoic acid. It was observed that plasticity at the neuronal synapses was altered with the excessive addition of UBE3A, thereby suggesting a possible link between retinoic acid signaling and UBE3A. Additionally, it also established the pivotal role of dosage compensation of the UBE3A gene in modulating cellular processes. Supplementing an RA antagonist or overexpressing UBE3A resulted in synaptic defects in ASD. However, these synaptic deficiencies were amended by the reconstitution of RA [148]. Therefore, in this process, reduced RA signaling was identified as a novel mechanism in ASD phenotypes linked to UBE3A dysregulation and was looked upon as a potential therapeutic intervention for ASD treatments. 

Phosphor-tyrosyl phosphatase activator [PTPA], known to stimulate protein phosphatase 2A (PP2A), was identified as a target of UBE3A [149]. Increased levels of neuronal PTPA and PP2A were observed in the transgenic AS-modelled mice. The pharmacological downregulation of PP2A or the hemizygous knockout of the Ptpa gene could rescue the defects of spine maturation in the neuron’s dendrites, indicating the relevance of the UBE3A–PTPA–PP2A pathway in AS pathogenesis.

Differential changes in UBE3A expression underlie many neurological phenotypes. One major limitation and challenge in the field that needs to be resolved is to identify UBE3A targets as disease-relevant substrates in the brain region. Another putative UBE3A target that has been previously published is Pbl/ECT2. It is described as a RhoA guanine nucleotide exchange factor (RhoA GEF), although its role in AS phenotypes has not been confirmed [98]. In a recent study, a contradictory report suggested that Ephexin5, a RhoA GEF, and a UBE3A target do not contribute to AS-related cortical and cerebellar phenotypes, including speech and communication defects, enhanced seizures, or defects in motor coordination [150,151]. A GABA transporter, known as GAT1, was observed to be upregulated in the UBE3A-deficient cerebellum. GAT1 has been reported to be yet another UBE3A target. Selective treatment with THIP, an extra-synaptic GABAA receptor agonist, treated electrophysiological and motor deficits [131]. The cytoskeleton-associated protein Arc, another target of UBE3A, is known to modulate the trafficking of AMPA receptors to the membrane [68]. Reduction of Arc levels, in a selective manner, improve recovery time after seizures without having much impact on speech vocalizations or motor coordination defects [151]. Irrespective of various results observed by different groups regarding Arc being a direct target of UBE3A [68,109,151,152], these data positively state that the lowering of Arc levels reduces AS-like symptoms. Additionally, many non-target proteins that are reported to be indirectly regulated by UBE3A are altered in AS-modelled mouse brains. Some of these proteins are α1-Na+/K+-ATPase and CaMKII [114,151,153,154]. The fly homolog of α1-Na+/K+-ATPase was identified as an interactor of dUBE3A, the *Drosophila* homolog of human UBE3A [155]. However, this result was non-reproducible in the mouse brain owing to the existence of redundant protein isoforms [99]. The deletion of specific genetic fragments of the CAMKII gene or other modifications of CaMKII phosphorylation sites were designed to modify the levels of the dysregulated proteins. This design underlined experimental strategies to treat certain behaviors, mostly about learning and memory paradigms [151,153,154]. Even with the reports that some of the clinical phenotypes were rectified, none of these research works succeeded in ameliorating all these clinical manifestations displayed by the AS transgenic mouse models. These results emphasized the fact that the link between the molecular understanding of these various altered dysregulated pathways and their treatment capacity is complex and needs to be further investigated in detail. One interesting fact about this data, highlighted during the detailed analysis of the wide spectrum of the varied research works, is that AS development can arise from several secondary and tertiary alterations downstream of UBE3A gene expression (Figure 3). Some of the proteins dysregulated in AS and/or interact with UBE3A have also been shown to be affected in other neurodevelopmental disorders, including ASDs (Table 3). All these data hint towards the existence of a phenotypic overlap between AS and other neurodevelopmental disorders. This link and the spectrum of overlap can be investigated further by studying the molecular connectivity in downstream cellular signaling pathways modulating the development of the nervous system.

There have been several obstacles in delineating the substrates of UBE3A causing neuronal defects. The standing hypothesis is that an alteration in UBE3A protein levels leads to changes in the levels of various downstream proteins, leading to various clinical phenotypes of AS. Proteins that are altered in AS are also associated with various other ASDs. These ASDs might not have any prior relationship with the changing levels of UBE3A, hinting towards the existence of a potential molecular connection, leading to the varied clinical phenotypes in various ASDs, (Figure 3). In this review, we have tried to list all the possible mammalian UBE3A substrates in Table 3 [156].

*GABRB3* is known to encode for the β3 subunit of the GABAA receptor, with high-level expression in the embryonic brain. Repressor-element-1-silencing transcription factor (REST) modulates neuronal gene expression in the embryonic brain. *GABRB3* is also expressed in high levels in the hippocampus of the adult brain, with low-level expression in the remaining region of the brain. There are three neurological disorders, Rett syndrome, Angelman syndrome, and autism, which show lower *GABRB3* and *UBE3A* gene expression, along with the manifestation of clinical phenotypes like intellectual disability and epilepsy. *GABRB3* is known to be strongly associated with epilepsy. The regulation of REST by UBE3A, which controls *GABRB3* expression, and MeCP2’s alteration of *UBE3A*, connect the neurological disorders like Rett, Angelman, and autism syndromes with epilepsy-like phenotypes, and emphasize the role of epigenetic mechanisms in causing epilepsy [157]. In another pivotal study, Santini et al. [91] exhibited increased levels of superoxide in the hippocampal brain region of AS mice and this biomolecular brain phenotype could be ameliorated by the administration of MitoQ 10-methanesulfonate (MitoQ), a mitochondria-related antioxidant. Additionally, MitoQ improved hippocampal synaptic plasticity defects and contextual fear memory deficits in the transgenic AS mouse model. All these experimental results implicate that mitochondria-associated oxidative stress leads to hippocampal pathophysiology in AS transgenic mouse models and pharmacological manipulations of mitochondrial ROS could aid AS patients. Even though Santini et al. [91] provide strong proof for the role of increased levels of mitochondrial superoxide in the hippocampus-associated synaptic and behavioral deficits in AS mice, the connection between E3 ligase E6-AP loss and elevated superoxide expression is yet to be delineated. There is speculation that one of the plausible mechanisms is the functioning of an abnormal electron transport chain (ETC), eventually leading to oxidative stress. A previous study by Su et al. [89] supports this hypothesis. According to Su et al., mitochondrial morphological defects in the AS mouse models correlated with a reduction in the complex III activity of the ETC. This leads us to think and conclude that there exists a possibility that E6-AP may regulate the expression of either machinery or modifiers of the ETC, and therefore the loss of E6-AP results in defective mitochondrial ETC.

**Table 3 diseases-12-00007-t003:** Possible UBE3A targets and their role in various ASDs: Table listing the possible targets of UBE3A in regulating various diseases encompassing the ASDs. AS: Angelman Syndrome; ASD: Autism Spectrum Disorder; TS: Tuberous sclerosis.

Substrates/ Interactors	Role of UBE3A in Modulating the Target	Disease	References
ANXA1	Ubiquitination	ASD	[158,159]
AR	Coactivator	ASD	[94,160]
Arc	Coactivator regulating transcription and Ubiquitination.	AS, Fragile X	[68,151,161]
CDKN1B	Ubiquitination	ASD	[97,162]
DLG1	Ubiquitination	ASD	[163,164]
Ephexin 5	Ubiquitination	Epilepsy	[69,165]
ESR2	Ubiquitination	Asperger Syndrome, ASD	[141,166]
Herc2	Ubiquitination	AS, ASD	[167,168,169]
mGluR5	Ubiquitination	AS, Fragile X	[170,171]
SOD1	Ubiquitination	ASD	[172,173]
TSC1	Ubiquitination	AS, TS, ASD	[139,174]
TSC2	Ubiquitination	AS, TS, ASD	[139,174,175]
UBE3A	Ubiquitination	AS, ASD	[176,177]

Defective mitochondrial ETC may result in oxidative stress. There is, however, a strong need for further research to delineate the mechanism by which E6-AP modulates the ETC to validate the above possibilities.

## 5. Potential Biomarkers Related to UBE3A

Duplications and triplications of 15q11.2–q13, are causal factors of autism spectrum disorder (ASD), intellectual disability (ID), delays in developmental processes, and epileptic seizures observed in Dup15q syndrome. Designing targeted therapies for varied forms of ASD and Dup15q syndrome requires biomarkers with the potential to aid in evaluating treatment response at the clinical and bio-molecular levels. Biomarkers can be used as potential stimulators for clinical trials and are computable measurement parameters of drug target efficacy, which, in turn, can influence assessments regarding the continuation of these clinical trials. Several biomarkers associated with UBE3A-mediated ASDs are listed in Table 4, and this table lists the methodology associated with biomarker detection in various research findings.

Electroencephalogram (EEG) forms one of the important characteristic quantifiable features of Dup15q syndrome that possibly reflects molecular pathological characteristics [64,67,178]. Frohlick and coworkers [67] quantified the EEG phenotype as spontaneous beta band (12–30 Hz) oscillations in children with Dup15q syndrome. These children were not taking benzodiazepines or other pharmaceutical drugs known to induce beta activity. The encouraging result of this study shows that the EEG phenotype in patients with Dup15q is a potential optimistic biomarker that might help in quantifying disease pathophysiology and drug–target efficacy, and shows optimistic potential for Dup15q syndrome therapies.

The effective application of the Dup15q syndrome biomarker requires an understanding of the underlying genes, their function, and Dup15q syndrome pathophysiology. There are several 15q genes associated with the disease progression, including UBE3A and a cluster of gamma-aminobutyric acid type-A (GABAA) receptors β3, α5, and γ3 subunit genes. To understand the role of UBE3A or the GABRB3/GABRA5/GABRG3 gene group’s role in mediating the beta EEG phenotype in Dup15q syndrome, Frohlick and coworkers [67] performed some studies. The prior characterization of the beta EEG phenotype showed strong, robust, and reproducible phenotypes in a larger Dup15q syndrome sample. In this study, the healthy and developing children cohort’s beta power was compared with the Dup15q syndrome children cohort with both interstitial and iso-dicentric duplications. The hypothesis is that the dysfunction of the GABAergic system is necessary to produce the beta EEG phenotype. To execute this, the GABAergic dysfunction in patients with Dup15q syndrome was compared to beta oscillations induced by midazolam [pharmacological GABAA modulator] in healthy adults. Additionally, they also evaluated whether UBE3A dysregulation was necessary for the beta EEG phenotype. For this, they compared two cases of paternal Dup15q syndrome to the reference cohort of children with Dup15q syndrome previously mentioned. Their studies were a step forward in efforts toward improving clinical trials in Dup15q syndrome since knowledge regarding the mechanistic underpinnings of the EEG biomarker paves the way for its application in pharmacological trials. These methodologies could therefore serve as an effective and potent quantitative methods for treatment response or drug-target engagement.

Studies have shown that abnormal sleep cycles occur in 40–80% of ASDs [179,180,181,182]. Poor sleep has been associated with enhanced autism severity, cognitive anomalies, and behavioral impairment [183,184]. Sleep physiology has its quantifiable structural attributes like spindles and slow wave sleep (SWS). Abnormal spindle number and morphology are associated with epilepsy, as well as with neurodevelopmental and neuropsychiatric disorders [185,186,187,188,189,190]. Abnormalities in SWS have been observed in neurodevelopmental disorders such as Rett syndrome ([191,192], in both genetic forms of ASD and non-syndromic forms of ASD [193]). One common methodology to quantify sleep physiology is an awake EEG.

Saravanapandian and coworkers [194] have used awake electroencephalography (EEG), as a methodology for screening for patients with Dup15q syndrome. The increased frequency of beta band waves (12–30 Hz) is an electrographic biomarker generated in this methodology which helps in distinguishing the patients with Dup15q from healthy developing and non-syndromic ASD children. The EEG signature matched with the pattern observed in patients placed on a benzodiazepine or allosteric modulators of GABAA receptor therapies. This validates that the EEG biomarker mimics abnormal GABAergic neuronal transmission. Additionally, in this research, the authors investigated the role of sleep physiology in Dup15q syndromic patients. Several metrics [beta band oscillations, spindle density, and percentage of SWS (slow wave sleeps)] were quantified from clinical EEG and polysomnogram (PSG) recordings of patients with Dup15q syndrome and age-matched controls. The results exhibited that Dup15q syndromic children showed aberrant sleep physiology with an elevated beta power, reduced spindle density, and reduced or absent SWS compared to age-matched controls. All these findings pointed towards the strong association between sleep EEG and cognition, identification, and the validation of potential pharmacological targets to improve sleep cycle defects and neurodevelopmental clinical manifestations in this syndrome. This study aided in establishing that abnormal sleep physiology might impede the healthy development of cognitive abilities. Additionally, this study also established that sleep EEG might serve as a robust computable target for behavioral and pharmacological interventions.

Godler and coworkers [5] evaluated differential alterations in the mRNA levels of UBE3A and SNORD116 located within the 15q11–q13 region between Chromosome 15-imprinting disorders (Prader–Willi (PWS), Angelman (AS) and chromosome 15 duplication (Dup15q) syndromes) and their subtypes in correlation with clinical phenotypes. In this study, 58 participants were affected with either PWS, AS, or Dup15q, and 20 healthy controls were included. Utilizing the advanced and sophisticated reverse transcription droplet digital polymerase chain reaction (PCR) technique, a semi-quantitative analysis of UBE3A and SNORD116 mRNA from peripheral blood mononuclear cells (PBMCs) was performed. The normalization of UBE3A and SNORD116 gene levels was performed concerning a panel of internal control genes (housekeeping constitutively active genes) using the geNorm approach. As a step forward, in this study, a functional evaluation of the intellectual development of the experimental subjects was also performed. The parameters set for this evaluation were set according to the standard and reliable guidelines laid by the Autism Diagnostic Observation Schedule-2nd Edition. The researchers found that the Dup15q group showed significantly more elevated levels of UBE3A mRNA than its healthy control counterparts (*p* < 0.001). The AS and Dup15q groups also had significantly elevated SNORD116 mRNA levels compared to controls (AS: *p* < 0.0001; Dup15q: *p* = 0.002). The mRNA levels of UBE3A and SNORD116 also exhibited a positive correlation with the functional intellectual development scores in the deletion AS subjects (*p* < 0.001) and the clinical autistic manifestations (*p* < 0.001) in the non-deletion PWS patients. These experimental results hinted towards the existence of novel molecular mechanisms underlying UBE3A and SNORD116 expression in PBMCs and brain-related processes associated with motor and language impairments and autism-like features in these disorders. Additionally, they hypothesized that distinct gene expression profiles in the PBMCs possibly will identify novel immune system-related biological processes in the brain. It is known that the specific expression of genes in the brain microglial cells modulates neuronal processes associated with the intellectual development of an individual. Any form of abnormal alteration in the expression levels of these genes in microglial cells leads to autism features in these disorders. Since direct gene expression profiling from brain neurons or glia is technically more cumbersome and challenging and can only be implemented in post mortem tissues, this study optimistically also facilitates the use of UBE3A and SNORD116 mRNA from PBMCs as potential molecular biomarkers for the clinical diagnosis of these neurodevelopmental disorders and readouts for testing the ameliorating effects of potent pharmacological drugs and gene therapies.

The neurophysiological underpinnings of UBE3A-mediated neurodevelopmental disorders are an unexplored arena and very little information is available for review. Nash and coworkers [75] generated a full-length UBE3A deletion in an AS rat model, with the aim and objective to elucidate novel mechanisms and therapeutic targets. This report demonstrated a novel finding that the UBE3A protein is detectable in an active form in the cerebrospinal fluid (CSF) of wild-type rats but is absent from the CSF of the AS rat. Within the rat’s hippocampal region, micro-dialysis was performed. It was found that the UBE3A protein was concentrated in the interstitial fluid of wild-type rat brains but was significantly absent in AS animals. These findings established that the UBE3A protein maintains its activity and is modulated in a way that strongly relies on its catalytic activity. Their findings proved that CSF can also act as a potential biomarker for UBE3A-mediated ASDs.

## 6. Potential Therapeutic Strategies against UBE3A-Mediated ASDs: Insights from Cellular, In Vivo, and Other Pre-Clinical Models

Therapeutic strategies for Angelman Syndrome (AS) were developed towards alleviating symptoms, including anti-seizure medications and physiotherapies in a traditional manner. Several approaches were proposed to develop mechanism-based therapies for AS.

**A. Approaches for the restoration of UBE3A expression:** Since a large fraction of AS cases are caused by the loss of UBE3A, as a first step, the restoration of UBE3A through direct gene therapy or through activating the paternal allele was employed [195]. The administration of recombinant adeno-associated virus (AAV) carrying the mouse UBE3A into the AS mouse’s hippocampus restored to localized UBE3A levels and ameliorated the animal’s learning and memory abilities [196]. The pitfall of this methodology is that the viral vectors showed a restricted localized distribution only in the hippocampus. The failure to distribute in other regions of the brain made this methodology unapt to ameliorate motor dysfunctions in these transgenic mouse models. Secondly, in this approach, a lack of precise control of UBE3A expression could also lead to unusually high UBE3A levels which could also act as a risk factor for ASD [9,108,197]. Therefore, while designing therapies based on UBE3A restoration, care should be taken to optimize the levels of UBE3A and introduce methods that can regulate the precise control of UBE3A gene expression.

**B. Approaches to Target UBE3A-ATS:** Focusing on the UBE3A-ATS [UBE3A anti-sense DNA] strand, which is transcribed as part of a larger transcript called *LNCAT* (large non-coding antisense transcript) at the *UBE3A* locus as the main target and reversing the silencing of paternal UBE3A in AS along with specific regional deletions or mutations in the maternal allele, is another therapeutic strategy [198,199]. Topoisomerase inhibitors, including topotecan, are known to inhibit long non-coding RNA UBE3A-ATS transcription in the nucleus, eventually aiding in the expression of UBE3A from the paternal allele [199,200]. Topotecan is known to bring about the transcriptional repression of many other synaptic genes and its usage can pose a concern due to its nonspecific pleiotropic effects. In summary, topoisomerase inhibitors modulating UBE3A regulation have provided optimistic results for future drug development strategies along with providing satisfying results of high efficacy. Antisense oligonucleotides (ASO), on the other hand, are an alternative and encouraging approach for gene therapies providing high specificity. However, the methods for the site-specific delivery of these antisense oligonucleotides remain challenging. UBE3A-ATS targeted by ASOs inhibit the silencing of the paternal UBE3A allele in neurons and improve certain cognitive defects in AS-modelled mice. The ASO failed to show any promising results in ameliorating certain other abnormal animal behaviors and recapitulating clinical manifestations of the disease [201]. The UBE3A-ATS-targeting oligonucleotides were designated as the orphan drug, under the Orphan Drug Act of 1983 by FDA, to aid in faster and more effective drug development. Dietary supplements could serve as an alternative strategy to reverse the silencing of the allele derived from the father. For example, methionine supplements were given to patients to enhance methylation, thereby suppressing the UBE3A ATS transcript [202,203]. Some of the clinical trials associated with the testing of methionine supplements showed no major difference in the clinical results between the control and the methionine supplement-treated groups. The promoter activation or inhibition of UBE3A-ATS using simulated transcription factors is another practicable strategy. The artificial transcription factor, TAT-S1, known to inhibit the UBE3A-ATS locus, has been shown to elevate UBE3A expression in the brains of AS transgenic mice [204].


**C. The Therapeutic Interventions of Downstream Effectors:**


Several strategies were explored to study the effect of therapeutic interventions on the downstream effectors of UBE3A in AS models. These comprised:
(i.)Lowering ARC levels: reducing ARC levels improved seizures in UBE3A-deficient mice without any significant difference in the motor dysfunction and altered ultrasonic vocalization defects [151].(ii.)The inhibition of calmodulin-dependent protein kinase II (CAMKII) phosphorylation: Levodopa reduces CAMKII phosphorylation and decreases the seizure intensity and motor deficits and alleviates hippocampal learning behavior and synaptic plasticity in the AS-modelled mice [153]. A clinical trial for the usage of Levodopa for AS is in the pipeline and clinically significant data is still awaited for further validation (NCT01281475).(iii.)Ampakines, known to act as modulators of AMPA receptors, have been shown to enhance BDNF release. This methodology improved hippocampus-related learning paradigms and proved to be useful in alleviating LTP in AS transgenic mouse models. Obstructing SK2 channels helped in improving LTP, learning, and memory behavior, and restored activity-dependent actin polymerization. Efforts to systemically inject an SK2 channel blocker have shown promising results in restoring animal fear-conditioning paradigms in the AS model [70,205,206].(iv.)The mTOR pathway has been implicated in the development of the brain and its synaptic plasticity. The over-activation of mTORC1 and inhibition of mTORC2 in UBE3A-deficient mice leads to motor abnormalities and LTP defects, anomalies in fear conditioning, and memory processes.

All these deficits and abnormalities can be ameliorated by the administration of the mTORC1 inhibitor Rapamycin and the mTORC2 activator A-443654 [207]. E6AP has been shown to modulate mTORC1 signaling by altering the levels of p18, a component of the Ragulator complex [208]. Rapamycin has proved to be beneficial for ASD associated with PTEN mutations. These results have been successfully validated in ASD-modelled mice [209,210].

A multi-dimensional approach is needed to target E3 ligases and should include detailed profiling of the substrates of E3 ligase. A comprehensive analysis of this enzyme’s structure and testing and validation of small molecule modulators are required. Proteasome inhibitors have been successfully used in clinics [211,212,213,214] and have surpassed major clinical trials. On the other hand, small molecules modulating E3 ligases are still in their early pre-clinical trial phases and need to be validated for safety and efficacy. The implementation of novel large platform screening technologies for the chemical compound library will pave the way for novel therapeutic findings. Activity-based probes, protein crystallography, and mass spectrometry need to be routinely incorporated to facilitate these drug discovery programs. Additionally, recently developed techniques, like **Pro**teolysis **Ta**rgeting **C**himeras (PROTACs), **S**pecific and **N**on-genetic **I**AP (inhibitor of apoptosis *protein*)-dependent **P**rotein **Er**asers (SNIPER) [215,216,217], will open roads for designing high-throughput, specific, and effective strategic plans to modulate the UPS machinery in human diseases.


**D. Other Therapeutic Interventions:**


Minocycline improves various clinically manifested human behaviors like cognitive dysfunction, schizophrenia, depression, and those associated with fragile X syndrome (FXS). Since the pathophysiological mechanisms between AS and FXS were found to overlap and commonly occur with shared clinical manifestations, a minocycline clinical trial was conducted in AS patients. The results of this pilot study were found to be promising and minocycline was able to exhibit clinical corrections in language disabilities, motor deficits, improper cognition, and adaptive behavior [218]. The minocycline clinical trials’ results in AS patients require more thorough investigations, as the previous cohort studies lacked control subjects, before making any conclusive claims. By using the antibiotic mithramycin [219], routinely known to be used for the treatment of tumors, Germain et al. could reinstate normal UBE3A mRNA levels in iPSC lines from idic15 patients. However, the use of mithramycin needs detailed investigation as mithramycin modulates the broad-spectrum expression of genes rather than specifically altering levels of UBE3A. In yet another study, Yi et al. showed that chronic treatment with drugs specifically modulating PKA, like forskolin and rolipram, can inhibit UBE3A action in the neurons. This paves new avenues for modifying upstream regulators of UBE3A to ameliorate clinical manifestations of autism [220]. Additionally, Roy et al., 2020 [221] conducted an unbiased drug screen for seizure suppressors in transgenic *Drosophila* models of Dup15q syndrome and found that serotoninergic and dopaminergic signaling pathway activation could be used as potential therapeutic interventions in suppressing epileptic seizures, one of the prominent clinical manifestations of Dup15q Syndrome. In this study, few compounds from the Prestwick Chemical Library were found to ameliorate seizure phenotypes. Most of these compounds altered and modified serotoninergic and dopaminergic signaling. The mechanism behind the seizure suppression occurred through the activation of serotonin receptor 5-HT1A. The Na+/K+ exchanger ATP alpha was modulated by the fly UBE3A in glial cells. Serotonin and dopaminergic signaling seem to regulate the above process, thereby leading to a decrease in seizures. Roy and coworkers, based on their pharmacological and genetic studies, emphasized and suggested the usage of 5-HT1A agonists in the therapy of UBE3ADup15q epilepsy. In yet another study, Cruz et al., 2021, [222] exhibited that insulin-like growth factor 2 (IGF-2) and mannose-6-phosphate (M6P), ligands of the cation-independent M6P/IGF-2 receptor (CIM6P/IGF-2R), improved the clinical deficits of AS modeled in mice. A reconstitution of IGF-2 or M6P in transgenic AS mice, subcutaneously, improved cognitive defect and contextual and recognition memories, rescued motor defects, improved working memory/flexibility, and also ameliorated marble-burying defects and reduced seizures in these mouse models.


**E. The Therapeutic time window for therapies for UBE3A-mediated ASDs:**


Reports suggest that the truncation of *UBE3A-ATS* genetically is more effective in treating phenotypic defects [223] than ASO treatment in AS mice. Various therapeutic implications are indicated by these scientific studies. Firstly, long-lasting unsilencing is required, and secondly, early prenatal UBE3A expression is important. The notion of the existence of a critical window period for therapeutic strategies for UBE3A-mediated ASDs needs to be kept in mind while designing and administrating these strategies. This notion of the existence of a critical window frame is supported by a recent study, which systematically examined the effects of restoring UBE3A expression during distinct neurodevelopmental windows. The study found that UBE3A restoration only during a very early time window, in the embryonic stage, fully treated AS-relevant phenotypes. UBE3A’s replacement during adolescence was semi-effective, while during adulthood it restored only hippocampal synaptic plasticity [224]. Thus, UBE3A restoration should be attempted as early as possible during post natal development. Yet, various other studies indicated that the optimal treatment window for restoring UBE3A expression in AS mice was shown to be around birth [224,225], and injections of ASO to restore *UBE3A* in vivo were implemented in neonatal (P1) *UBE3A^m−/p+^* (AS) mice [110]. Additionally, in scientific studies conducted by Roy and coworkers [226], the pharmacological intervention of various drugs in the large-scale Prestwick Drug Library screening was administered in the embryonic stage of flies and continued throughout their development until adulthood.

## 7. Conclusions

Numerous genes have been associated with various forms of clinical manifestations in ASDs. This review article has been aimed at highlighting one important gene, UBE3A, and its protein product, in causing ASDs. We have provided a comprehensive review of how the UBE3A molecule is involved in ASDs. We have aimed at highlighting its structure and function, the mechanisms underlying UBE3A-mediated ASDs, how it can be used as a potential biomarker, and methodologies that have been adopted to use it as a therapeutic intervention. Additionally, we have discussed the pitfalls of some therapeutic interventions and the key concerns that must be kept in mind while designing therapies in the future. We hope that this comprehensive review may inspire all those new to the science of ASD and will aid in the beginning of novel research and help additionally in making steady developments in the field of UBE3A’s role in triggering ASDs. We hope that the subsequent findings along with the already existing vast ocean of knowledge will be used in devising efficient tools for diagnosis and therapy. Existing data available from worldwide research indicates the importance of UBE3A in regulating pathogenesis and clinical symptoms in ASD patients. This amazing and important molecule can potentially be utilized as one of the key diagnostic biomarkers for ASDs. However, more research should be performed on therapeutic interventions for a robust, potent, effective strategy with minimal costs and low side effects. Designing specific strategies to regulate the levels of UBE3A gene expression in the CNS to restore its optimum levels is challenging for scientists. The value of the extensive research work accomplished so far has aided and will always positively help in elucidating innovative and unique molecular mechanisms and underlying pathways regulating UBE3A-mediated ASDs. Research studies conducted so far are just positive forward steps toward understanding the mechanisms of ASDs and designing more cost-effective and potent UBE3A-based diagnostic tools and therapies for the future.

## Figures and Tables

**Figure 1 diseases-12-00007-f001:**
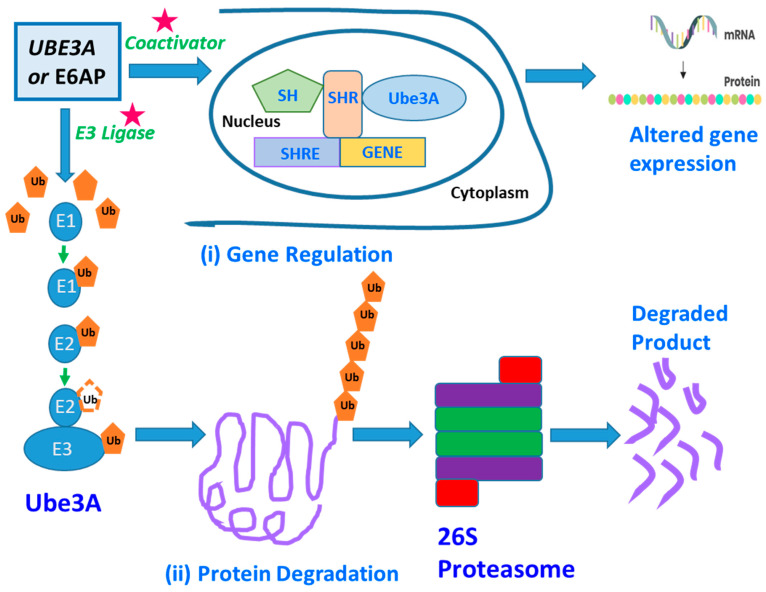
This figure represents the functional role of UBE3A or E6AP: UBE3A selectively targets various Ub-tagged proteins in the cytoplasm through the Ubiquitin–proteasome pathway and this degradation pathway regulates various basic neuronal functions (**ii**). Alternatively, UBE3A could modulate various neuronal genes’ expressions in the nucleus, owing to its ability to function as a co-activator in the mRNA transcription pathway. Altered mRNA levels in the cytoplasm affect protein levels (**i**). Over-activation or loss of UBE3A’s activity could alter the function and plasticity of the synapse. Altered synaptic function and plasticity could potentially serve as one of the fundamental causes of many behavioral defects and clinical manifestations observed in autism and AS. Abbreviations: SH stands for steroid hormone; SHR stands for steroid hormone receptor; SHRE stands for steroid hormone receptor element; Ub stands for Ubiquitin; E1 stands for Ubiquitin-activating enzyme; E2 stands for Ubiquitin-conjugating enzyme; and E3 for Ubiquitin ligase.

**Figure 2 diseases-12-00007-f002:**
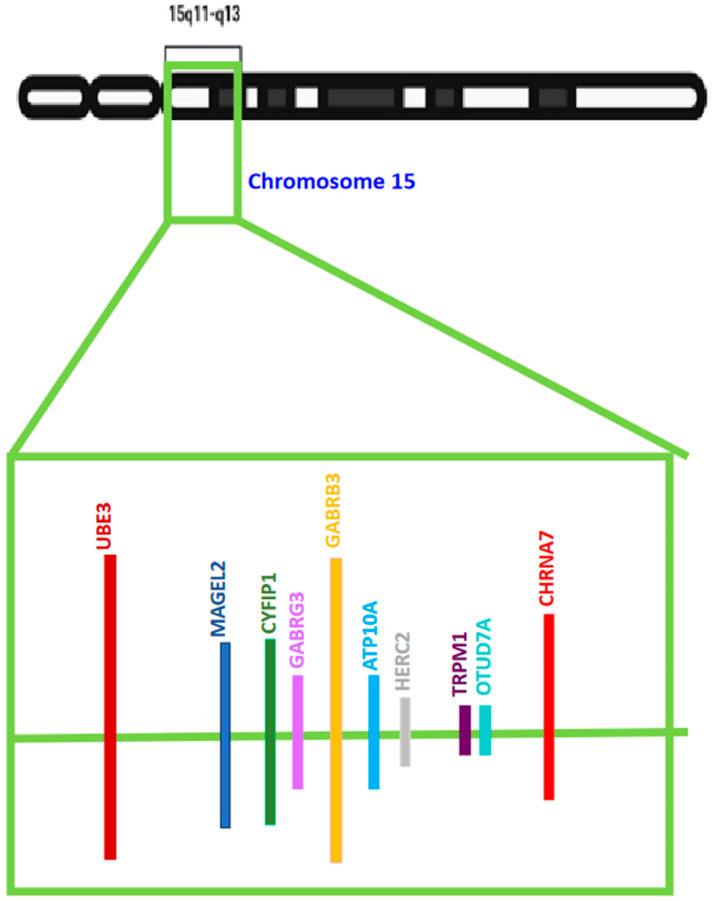
This figure displays the schematic showing some of the important genes for ASD situated in the Chr15q11–q13 segment. Even though the position and the size of genes have not been appropriately scaled, we have represented several genes that span the 15q11–q13 chromosome segment. Our gene of interest, UBE3A, is one of the candidates with a genetic location between 25,333,728 and 25,346,031.

**Figure 3 diseases-12-00007-f003:**
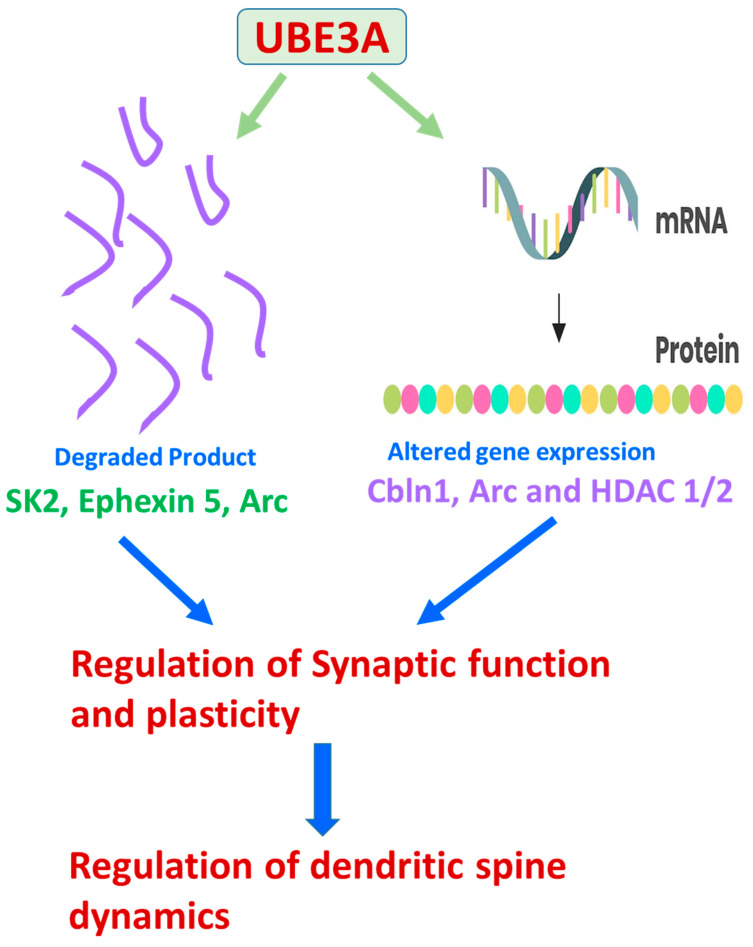
This figure represents UBE3A’s functional role in modulating synaptic structure, transmission, and neuronal plasticity: UBE3A can target specifically synaptic proteins like Ephexin5, SK2 Arc, etc., via proteolytic cleavage by the Ubiquitin–Proteosome pathway. UBE3A can regulate the expression levels of numerous genes like Cbln1, Arc, HDAC1, and 2, etc., regulating synaptic dysfunction by playing an alternative role as a co-activator. Overactivation or functional loss of UBE3A could alter neuronal function and plasticity by altering the levels of various target proteins. Henceforth, this could serve as the underlying mechanism of many behavioral defects and other clinical symptoms observed in both AS and autism.

**Figure 4 diseases-12-00007-f004:**
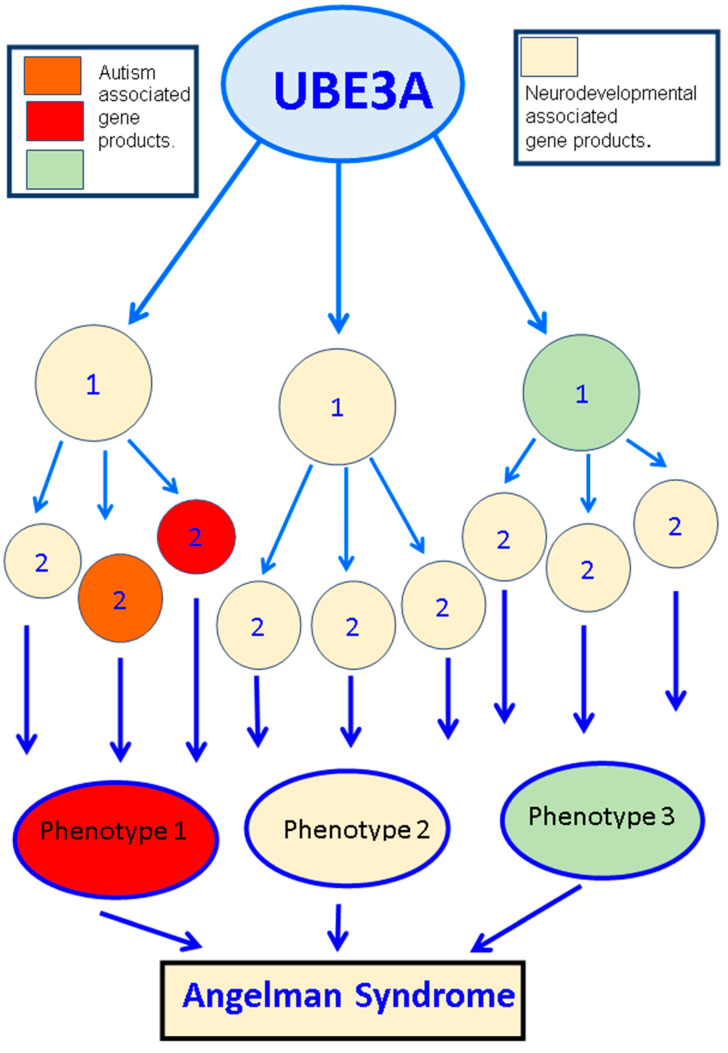
This figure is a schematic flow chart of UBE3A targets in Angelman Syndrome (AS): Angelman Syndrome is caused by the deficiency of maternal UBE3A protein expression, resulting in the deficiency of downstream substrates. Item 1 represents the direct interactors/targets and item 2 represents the indirect targets. Gene products associated with neurodevelopmental processes are light-orange colored and the ones associated with autism spectrum disorders are green, dark orange, and red colored. The secondary changes associated with the alteration in the levels of gene products can be at the level of protein–protein interactions, cell-biological, and electrophysiological properties of the neurons. The orchestrated union of both secondary and downstream changes produces phenotypic characteristics of Angelman Syndrome. It is hypothesized or speculated that some proteins known to be dysregulated in ASD are either first-level (primary) or second-level (secondary) interactors in this UBE3A-dependent pathway and are known to produce overlapping phenotypic characteristics in AS and ASDs, hinting towards a potential overlapping molecular change in AS and ASDs.

**Table 1 diseases-12-00007-t001:** (**A**) Table representing the genes localized in the Chromosome 15q region and identified in various clinical studies that could potentially be involved in ASDs. (**B**) Table representing the genes, localized in the Chromosome 15q region, which were identified in various in vivo studies and have the potential to be involved in ASDs. (**C**) Table representing the genes, localized in the Chromosome 15q region, which were identified in various in vitro studies and have the potential to be involved in ASDs.

**(A)**							
**S.No.**	**Case Study Group**	**Author Details**	**Analytical Methodology**	**Gene Location**	**Genes Modulated**	**Type of Mutation**	**Reference**
1	100 families, of which 42 affected sibling-pair families and 58 trios were tested.	Nurmi et al., 2003.	Genotyping of SNPs was performed either by using allele-specific oligonucleotide hybridization (ASO), pyrosequencing analysis (PSQ), or fluorescent polarization-template, directed-dye terminator incorporation assay [FP-TD1].	The maternal expression domain of 15q11–q13 and ICRs.	Two SNPs are located within the gene ATP10C.	SNPs	[2]
2	166 ASD patients and 416 healthy individuals.	Kato et al., 2008	SNP analysis based on sequencing technology.	Maternal expression 15q11– q13 domain.	Marker haplotype located in ATP10C gene.	SNPs	[3]
3	20 ASD patients and controls.	Gregory et al., 2009	Control patients and 20 ASD patients. Defects in oxytocin function were found in the ASD patients.	15q11.2	Receptor for Oxytocin [OXTR]	Hypermethylation or excessive methylation of the gene promoter, with a reduced mRNA expression.	[4]
4	58 participants (affected with a C15-imprinting disorder), 20 normal control patients.	Baker et al., 2020	Reverse transcription ddPCR (droplet digital polymerase chain reaction) was used for gene expression assay	15q11–q13	**UBE3A**, SNORD116.	Duplication	[5]
5	2 postmortem brain tissue samples.	Hogart et al., 2009	RT-PCR followed by Southern blot, bisulfite sequencing, and fluorescence in situ hybridization.	15q11–q13	Paternally expressed transcripts of SNRPN, NDN, HBII85, and HBII52 showed deficiencies. Increased DNA methylation of the 15q11–q13-imprinted control region (ICR) was observed.	Duplication	[6]
6	146 subjects with autism/ASD.	Burnside et al., 2001	Cytogenomic microarray analytical tools were implemented.	Proximal 15q11.2.		Deletion or duplication of the BP1-BP2 region on proximal 15q.	[7]
7	15-year-old female case report study.	Han, 2021	Array-comparative genomic hybridization (aCGH) of chromosome 15q11.2–q13.1 was performed.	Chromosome 15q11.2–q13.1.	Maternally expressed gene product in the critical region of PWS/AS.	Interstitial (int) dup15q	[8]
8	146 parents, 79 pro-bands, and seven healthy siblings amongst a total of 232 subjects.	Guffanti et al., 2011	Low-density genotyping followed by secondary analysis using a densely spaced set of tag-SNPs spanning the target area.	15q11–q13	**UBE3A** and ATP10A.	An association of autistic phenotypes with an SNP located in the intergenic region between UBE3A and ATP10A was observed.	[9]
9	267 patients showing clinical manifestations like intellectual disability, autism, epilepsy, and congenital defects.	Iourov et al., 2015	SNP chromosomal microarray.	Long continuous stretches of homozygous (LCSH) regions of 7q21.3, 7q31.2, 11p15.5, and 15p11.2.	**UBE3A**	Single nucleotide polymorph-hism (SNP).	[10]
10	504 children with ASD and intellectual disability defects.	Iourov et al., 2019	SNP-based chromosomal microarray.	Long continuous stretches of homozygous in situ (LCSH) regions of 7q21.3, 7q31.2, 11p15.5, and 15p11.2.	**UBE3A**, MAGEL2.	Single nucleotide polymorphism (SNP).	[11]
11	297 affected children with language impairment).	Nudel et al., 2014	GWAS [Genome wide association] analysis, using a genotyping array.	Chromosome 14q12, Chromosome 5p13.	Several genes including the NOP9 gene in the paternal region; chromosome 5 regions fall between the PTGER4 and DAB2 genes	Paternal parent-of-origin effects on chromosome 14q12; suggestive maternal parent-of-origin effects on chromosome 5p13.	[12]
12	6 chimpanzees and 25 human brain tissue samples were analyzed.	Schneider et al., 2014	Methylated DNA immunoprecipitation (MeDIP) combined with DNA methylation arrays.		CNTNAP2	Protein-coding splice variant CNTNAP2-201 is 1.6 times upregulated in the human cortex, with widespread cortical DNA methylation changes in CNTNAP2.	[13]
13	43 ASD patients and 38 controls (a total of 223 post-mortem tissue samples isolated from three brain regions [prefrontal cortex, the temporal cortex, and the cerebellum]).	Wong et al., 2019	Genome-wide DNA methylation profiling.	Chromosome 15q (dup15q).	**UBE3A**, ATP10A.	Duplication and DNA methylation.	[14]
14	A single male patient case report study.	Wu et al., 2009	Array comparative genome hybridization (aCGH).	Chromosome 15.		Chromosome 15 duplication arising from a 3:1 segregation error of a paternally derived translocation between chromosome 15q13.2 and chromosome 9q34.12, which led to trisomy of chromosome 15p-q13.2 and 9q34.12-q.	[15]
15	993 families with 896 sibling pairs from the AGRE (Autism Genetic Resource Exchange) and 223 families (174 affected sibling pairs) from the NIMH (National Institute of Mental Health) Autism Genetics initiative were included.	Fradin et al., 2010	SNP genotyping, followed by a genome-wide linkage scan using parametric and non-parametric linkage analysis.	Chromosome 15q.	CLOCK gene.	SNPs	[16]
16	522 patients.	Depien-ne et al., 2009.	Multiplex ligation-dependent probe amplification (MLPA).	Chromosome 15q11–q13 region.	**UBE3A**, GABRB3.	Deletions, duplications, and methylation abnormalities.	[17]
17	82 patients with autism.	Curran et al., 2005.	Extended transmission disequilibrium test (ETDT).	Chromosome 15 (q11–q13).	GABRB3	Microsatellite markers, hemizygous deletion.	[18]
18	18 patients with ASD and 15 controls; post mortem tissue samples were analyzed.	Ben-David et al., 2014.	Genome-wide survey of allele expression imbalance (AEI).	Chromosome 15q11–q13.	**UBE3A**	SNPs	[19]
19	Postmortem human brain tissue (8 from dup15q, 10 patients with idiopathic autism, and 21 controls)	Scoles et al., 2011	Quantitative RT-PCR and Western blot analyses.	Chromosome 15q11–q13.	**UBE3A**	Duplication	[20]
20	A single female patient case study.	Noor et al., 2015.	Array comparative genomic hybridization (aCGH) analysis.	15q11.2 duplication encompassing only the UBE3A gene.	**UBE3A**	Maternally inherited 129 Kb duplication in chromosome region 15q11.2 encompassing only the UBE3A gene.	[21]
21	10 autistic patients.	Veenst-ra et al., 1999	Mutation screening of the UBE3A/E6-AP gene in autistic disorder.	15q11–q13.	**UBE3A**	Coding region and a putative promoter region.	[16]
22	6-month-old girl case report study.	Kitsiou-Tzeli et al., 2010	Array comparative genomic hybridization (aCGH) analysis.	15q11–q13.		Duplication of maternal origin of the 15q11.2–q14 PWS/AS region.	[22]
23	12 family members, in a family, tested positive for the 15q11q13 duplication.	Piard et al., 2010	Fluorescence in in situ hybridization (FISH), PCR analysis of microsatellite markers, array-comparative genomic hybridization analysis (Array-CGH) and semi-quantitative methylation-sensitive PCR.	Chromosome 15q11–q13.	CYFIP1, MAGEL2, SNRPN, **UBE3A**, and GABRB3.	Interstitial duplication of chromosome 15q11–q13.	[23]
**(B)**							
**S.No.**	**Details of the Authors**	**Model Organism Being Investigated**	**Position on the Chromosome**	**Genes Involved**	**Type of Mutation**	**Role of the Gene in the Cellular Process**	**Reference** **in Pubmed**
1	Leung and coworkers, 2009	C57BL/6 mice	15q11–q13 snoRNA HBII-85 locus.	snoRNA	Locus of UBE3A gene with the mutation for transcriptionally regulated chromatin decondensation or unfolding.	Gene transcription, neurodevelopmental processes and long-term memory.	[24]
2	Samaco and coworkers, 2005	Mouse null for Mecp2. Brain region was investigated.	Maternal 15q11–q13 or UBE3A deficiency.	Mecp2, **UBE3A** and Gabrb3.	Mutation in Mecp2.	Synaptic excitatory or inhibitory homeostasis maintenance.	[25]
3	Hogart and coworkers, 2008	Mouse	15q11–q13	Atp10a/ATP10A.	Imprinting of the maternally expressed gene Atp10a/ATP10A.		[26]
4	Powell and coworkers, 2013	Mice	15q11–q13	**UBE3A**	Mice with Snord116 deletion exhibit increased UBE3A-ATS levels.		[27]
5	Jiang and coworkers, 2010	C57BL/6 mouse	UBE3A Gabrb3 deletions of 15q11–q13	**UBE3A**, Gabrb3, Atp10a	Maternal deletion from the UBE3A to the Gabrb3 gene region.		[28]
6	Chibuk and coworkers, 2001	C57BL/6JEi and M. spretus SPRET/Ei DNA	Chromo-some 15q.	NDNL2			[29]
**(C)**							
**S.No.**	**Author Details**		**Investigation** **Cellular Models**	**Position**	**Genes Involved**	**Mutation Type**	**Reference**
1	Thatcher et al., 2005		SH-SY5Y neuroblastoma cells.	15q11–q13	SNURF/SNRPN	Imprinting of 15q11–q13.	[30]
2	Lopez et al., 2017		Human neuroblastoma cells, including parental SH-SY5Y and the SH(15M) cellular model.	15q11–q13	**UBE3A**	Duplication of the 15q11–q13 locus.	[31]
3	Dunaway et al., 2016		Dup15q cell model (SH15M) and parental SH-SY5Y (SH) cell lines.	15q11.2–q13.3	**UBE3A**	15q11.2–q13.3 maternal duplication.	[32]

**Table 2 diseases-12-00007-t002:** Table representing the frequency of various genes detected in numerous clinical, in vitro, and in vivo studies associated with ASDs. Table 2 has been prepared by listing the number of times various genes have been detected in the studies listed in clinical, in vitro, and in vivo studies referred to in Table 1A–C.

S.No.	GENES	No. of Times Genes Detected in Various Clinical, In Vitro and In Vivo Studies. [Total 31 Studies]
**1**	**ATP10C**	2
**2**	**OXTR**	1
**3**	**UBE3A**	**16**
**4**	**SNORD116**	1
**5**	**SNRPN**	1
**6**	**NDN**	1
**7**	**HB1185**	1
**8**	**HB1152**	1
**9**	**ATP10A**	4
**10**	**MAGEL2**	2
**11**	**NOP9**	1
**12**	**PTGER4**	1
**13**	**DAB2**	1
**14**	**CNTNAP2**	1
**15**	**CLOCK**	1
**16**	**GABRB3**	5
**17**	**CYF1P1**	1
**18**	**SNRPN/SNURF**	1
**19**	**SnoRNA**	1
**20**	**Mecp2**	1
**21**	**NDNL2**	1

**Table 4 diseases-12-00007-t004:** Biomarkers associated with UBE3A-mediated ASDs: Table listing the biomarkers and the methodology associated with the biomarker detection in various research findings.

Biomarker	Methodology	Reference
Beta EEG	Spontaneous EEG recordings.	[4]
Awake electroencephalography (EEG)	Spindle density, beta power, and slow-wave sleep (SWS) percentage in sleep EEG recordings as quantitative parameters for measurements of clinical manifestation of ASD symptoms.	[78]
UBE3A and SNORD116 mRNA levels	Semi-quantitative techniques like reverse transcription droplet digital polymerase chain reaction (PCR) were used for UBE3A and SNORD116mRNA analysis from peripheral blood mononuclear cells (PBMCs). The data were normalized to a panel of internal control genes using the geNorm approach.	[79]
UBE3A protein	Micro-dialysis of CSF fluid located in the rat hippocampus.	[5]

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
