# Peer review of "UBE3A: The Role in Autism Spectrum Disorders (ASDs) and a Potential Candidate for Biomarker Studies and Designing Therapeutic Strategies"

_diseases, 2023, doi:10.3390/diseases12010007_

Round 1

Reviewer 1 Report (Previous Reviewer 1)

Comments and Suggestions for Authors

The authors of the manuscript titled "UBE3A: Role in Autism Spectrum Disorders (ASDs), a potential candidate for biomarker studies and designing therapeutic strategies" have improved their original manuscript. Though the novelty of the manuscript looks still quite poor, I greatly appreciated their efforts to address the previous concerns. Nonetheless, prior to publication adjustments are still required. The main text would benefit from a substantial trimming. Being the journal "Diseases" I would strongly suggest skipping all the unnecessary details regarding the molecular structure, splicing variants, etc... They are interesting but may be not here. Conversely, within the Section 6. "Therapeutic Strategies Against UBE3A-mediated ASDs" the issue of "Therapeutic" would greatly deserve some clarification. Most of the studies so far undertaken are quite far away from clinics and restricted either to cellular or at the very best to a preclinical model/s. Thus, conversly from that of the manuscript, the title of the Section 6 looks a little bit misleading. Currently, most of the approaches are promising and potentially useful for humans, but clinic is not just around the corner.

Comments on the Quality of English Language

The English language to many extents is still quite verbose, thus benefiting from a serious trimming throughout the main text

Author Response

  1. The authors of the manuscript titled "UBE3A: Role in Autism Spectrum Disorders (ASDs), a potential candidate for biomarker studies and designing therapeutic strategies" have improved their original manuscript. Though the novelty of the manuscript still looks quite poor, I greatly appreciated their efforts to address the previous concerns. Nonetheless, prior to publication adjustments are still required. The main text would benefit from a substantial trimming. Being the journal “Diseases”, I would strongly suggest skipping all the unnecessary details regarding the molecular structure, splicing variants, etc... They are interesting but maybe not here.

AUTHOR’S RESPONSE:

Since this is a comprehensive review of the Ube3A gene, we would like to keep these contents. Even if the article is being published by Diseases, it will be read by readers from varied backgrounds who will be interested in the translational link between Ube3A and ASDs. This will include clinicians, basic molecular biologists, biochemists, neuroscientists, and others. We would like to keep the spectrum of this article catering to readers from a wide variety of backgrounds and thereby would not like to remove the structure and function section, details of the molecular structure, and splicing variants.

  1. Conversely, within Section 6. "Therapeutic Strategies Against UBE3A-mediated ASDs" the issue of "Therapeutic" would greatly deserve some clarification. Most of the studies so far undertaken are quite far away from clinics and restricted either to cellular or at the very best to a preclinical model/s. Thus, conversely, from that of the manuscript, the title of Section 6 looks a little bit misleading. Currently, most of the approaches are promising and potentially useful for humans, but the clinic is not just around the corner.

AUTHORS RESPONSE:

We have changed the heading to:

Potential therapeutic strategies against UBE3A-mediated ASDs: insight from cellular, in vivo and other pre-clinical models.

Reviewer 2 Report (Previous Reviewer 3)

Comments and Suggestions for Authors

In this revised version, authors have duly addressed most of the issues in the original version.

Papers newly cited in this version should be added into References and renumbered in the text.

Author Response

In this revised version, the authors have duly addressed most of the issues in the original version. Papers newly cited in this version should be added to References and renumbered in the text.

AUTHOR’S RESPONSE:

We have added all the new citations to the text and the numbers have been adjusted by Endnote software.

Round 2

Reviewer 1 Report (Previous Reviewer 1)

Comments and Suggestions for Authors

The authors partly addressed the concerns raised by the reviewer and to some extent, it has been appreciated. Nonetheless, the manuscript remains quite poor in terms of novelty and, to many extents, too verbose.

Comments on the Quality of English Language

The English language would benefit from a thinning out. The style requires some amendments for example by replacing colloquial terms with scientific ones (i.e., line 65 "...which got picked up in...", identified/ascertained/detected would be better, etc...). 

Author Response

The authors partly addressed the concerns raised by the reviewer and to some extent, it has been appreciated. Nonetheless, the manuscript remains quite poor in terms of novelty and, to many extents, too verbose.

AUTHOR’S COMMENTS:

This article is novel as this cites updated literature on the recent findings of Ube3A's role in ASD. It is also a comprehensive review describing the molecular mechanisms underlying Ube3A-mediated ASDs, listing the invivo and invitro translational models of the disease, potential biomarker studies, and various lab-based and clinical therapeutic findings. So far, people have covered reviews separately on each of the topics and featuring other genes of ASD. Our review is novel as it lists one of the main genes associated with ASD. We have already mentioned the reason and the literature support for choosing this gene.

  1. Comments on the Quality of English Language

The English language would benefit from a thinning out. The style requires some amendments, for example by replacing colloquial terms with scientific ones (i.e., line 65 "...which got picked up in...", identified/ascertained/detected would be better, etc...).

AUTHOR’S COMMENTS:

We have made changes to line 65 and replaced colloquial words with more scientific words.

This manuscript is a resubmission of an earlier submission. The following is a list of the peer review reports and author responses from that submission.

Round 1

Reviewer 1 Report

Comments and Suggestions for Authors Dysfunctions of E3 Ubiquitin ligases, the enzymes responsible for tagging substrate proteins with Ubiquitin (Ub) moieties -either as monomer or as Ub-chains-, are implied in a plethora of disorders, including neurological. The manuscript review titled "Ube3A: Role in Autism Spectrum Disorders (ASDs), a potential candidate for biomarker studies and designing therapeutic strategies" by Roi B. and colleagues attempt to review the Autism Spectrum Disorders (ASDs) and their association with an E3 Ub-ligase, namely Ube3A. Among the hundreds of E3 Ub-ligases encoded by the human genome, when compared to the RING- the Hect-subfamily is very small in terms of members, but they are peculiar in terms of enzymatic activity. Indeed, they catalyze substrate ubiquitination in a two-step reaction, where initially they accept the activated Ub from the E2 in a transthiolation reaction on their unique catalytic cysteine, and then the Ub moiety is transferred directly from the E3 to a lysine on the target substrate. Honestly, in reading the manuscript I felt a little awkward. Besides lacking novelty (see just a couple of examples https://doi.org/10.3389/fnmol.2018.00448, https://doi.org/10.3389/fnins.2015.00322, https://doi.org/10.3389/fnmol.2018.00476) the manuscript looks drafted in a hasty way. Reading it appears quite hard because to many extents it is extremely redundant and with pointless parts. Furthermore, several inaccuracies and conceptual oversights are present throughout the manuscript. Basically, if the manuscript would bear some novelties it should be re-written from scratch.

Comments on the Quality of English Language

The English language requires profound editing because to many extents looks cumbersome.

Reviewer 2 Report

Comments and Suggestions for Authors

Thank you for the opportunity to review this comprehensive manuscript reviewing current knowledge about the UBE3A gene and its involvement in neurodevelopmental disorders. The authors’ aims, to review current literature about UBE3A’s involvement in neurodevelopmental disorders, its structure and function, molecular mechanisms, related biomarkers, and therapeutic strategies targeting UBE3A are ambitious, and it is useful for the field to have reviews that compile this information. However, this reviewer finds the current version of the manuscript to be very lengthy and repetitive, making it challenging to read and retain. The repetitive nature of the text and sometimes the order of material presented also obscures presentation of valuable information about UBE3A. Toward improving the impact of this review, I have listed several specific comments below, but emphasize that the full text would benefit from editing to remove redundant and superfluous text.

(1) For Tables 1A, 1B, and 1C, how were the listed studies identified? Which search terms were used in which database(s), and what criteria were applied to determine whether a study would be included in these tables? 

(2) The first paragraph describing Angelman Syndrome (lines 139-149) does not describe any symptomatic overlap with ASD, which seems a missed opportunity given the title and stated focus of the review (association and involvement of UBE3A in ASDs).

(3) Figure 1 legend describes synaptic involvement of UBE3A, but the figure does not include anything about the synapse. Consider removing this text about the synapse from this legend, or add an illustration of the synapse to Figure 1.

(4) Section 2 on “UBE3A Structure and function” requires substantial editing to remove repetitive and redundant text. For example, frequently repeated information includes: E6AP is the protein product of UBE3A, that genetic imprinting regulates UBE3A’s expression in brain, that only maternal gene copies are expressed in brain, that altered expression levels of UBE3A are associated with Angelman Syndrome and other diagnoses, that UBE3A is located on chromosome 15 and specifically within the 15q11-q13 region, etc. 

Structure and function of UBE3A are described approximately in lines 226-231, 232-243, 284-295, 315-318, and text about model systems in lines 319-355. Much of the other text in this section can likely be removed or relocated.

Additional information about UBE3A structure is included in Section 3, lines 368-381, which describes different UBE3A isoforms. This information would likely fit better if moved up to Section 2.

Lines 251-258 and lines 390-399 include additional phenotypic description of AS. This information would fit better earlier in the manuscript, where AS is first described (lines 139-149).

(5) This reviewer questions the necessity of Section 3 on “Regional association of UBE3A in ASD”, as much of the text in this section is repeated from previous sections, and/or is material that could fit better in other sections.

(6) Currently, the content of Figure 2 could be equally well conveyed by a table listing the genes of interest. To improve the utility of Figure 2, it would be useful to display all genes in the chr15q11-q13 region and to use the colors and lengths of lines noting each gene to convey additional information (number of studies supporting involvement in neurodevelopmental disorders? Imprinting regions? Expression level in the brain? Other?). 

(7) In Section 4, “Molecular mechanisms underlying UBE3A-mediated neuronal dysfunction in ASD”, it would be very helpful to define what is meant by “Ube3a target” for each example presented. Does “target” refer to a gene that is ubiquitinated by Ube3a, or a gene that is transcriptionally regulated by Ube3a’s co-activator function?

(8) Long paragraph at lines 543-583 includes description of several UBE3A targets that are described previously (e.g. Ephexin V, Arc). Please check if this is redundant to information presented previously, or clarify what is new about the content presented in this paragraph.

(9) Figure 4 is a bit confusing. What are some example phenotypes represented in this diagram? Is there a place for ASD in this diagram, either alongside, or overlapping with Angelman Syndrome?

(10) Table 3, please define what is meant by “substrate” – are these UBE3A ubiquitination targets or co-activation/transcription targets? 

(11) Section 5, please clarify (or clearly emphasize) if UBE3A mRNA or protein measures in blood or CSF have been reported as useful biomarkers of only AS, PWS, Dup15q, or if there is also published data supporting their use as biomarkers of idiopathic ASD? 

Minor comments

(1) Lines 51-52, clarify that “several scientific analyses and research studies” refers to studies specifically of the 15q11-q13 region 

(2) Tables 1A, 1B, 1C include some unusual hyphenations that should be corrected, e.g. “chromos-ome”, “hyperme-thylation, “express-ion”, “polymorph-hism” (there may be others)

(3) Lines 78-79 refer to a list of genes in a resource called the “Autism Database” but the cited references 30-32 do not correspond to such a database. 

(4) Lines 82-83, “intellectual disability” is currently accepted term instead of “mental retardation”

(5) Lines 121-122, imprinted genes are not truly haploid, but are “functionally haploid” (term used by the cited reference)

(6) Line 179 refers to “any individual with this syndrome” but it is not clear which of the above-mentioned syndromes this comment refers to. Angelman Syndrome? Please clarify.

(7) Paragraph at lines 181-189 is out of place. This paragraph reads like a concluding paragraph for the full review. Please delete or move.

(8) Line 591 should reference Table 3, not Table 1.

(9) Consider revising title of Section 5 to “Potential biomarkers related to UBE3A”, since EEG measures can indicate UBE3A-related disorders, but do not involve direct measurement of UBE3A

Reviewer 3 Report

Comments and Suggestions for Authors

This article by Roy et al provides an extensive review on the information about the roles of Ube3A in Angelman syndrome (AS), duplication 15q syndrome and other conditions of autism spectrum disorders (ASDs).

In general, the manuscript is well written. However, authors should address the following issues:

1.    Why is awake electroencephalography a biomarker for Ube mediated ASDs? Authors should explain the biological background.

2.    What is the therapeutic time window for the therapies of Ube3A-mediated ASDs?

3.    Despite the description E6AP is a UBE3A product (Figure 1), “E6AP” is not shown in Figure1.

4.    Figures 3 (lower half) and 4 are vague and not very informative.

5.    There are redundant descriptions about the clinical features and genetic causes of AS: Lines 139-149, 251-259, and 390-399.

6.    Ephexin5 is mentioned twice in Section 4 (Lines 441-454 and 548-552).

7.    There are numerous unnecessary commas, especially in Sections 4 and 5 (Lines 508, 580, 642, 663, 671, 676, 680, 681, 681, 695, 707, 718 and 817).

8.    Some words and phrases are either incomplete or incorrect: the pivotal roles this gene plays in regulating synaptic plasticity and functions (Lines 273-274); However (Line 492); PBMCs associated expression of these genes (Lines 722-723); the Orphan Drug Designation The Orphan Drug Act of 1983 (Lines 780-781).

9.    There are several typographic and other errors: Table 4 -> Table 2 (Line 55); Unnecessary blue marker after “UBE3A” (Table1(A), No. 20); Ube3A -> UBE3A (Lines 172 and 293); Inadequate indent (Line 439).

10. The use of abbreviations is inconsistent and sloppy: Angelman syndrome (AS) (Lines 80 and 117); retinoic acid (Line 529) versus RA (Lines 531 and 534); idic15 (Line 151) versus idic(15) (Line 842).

Comments on the Quality of English Language

English language is well, however, this manuscript requires further extensive editing.

Reviewer 4 Report

Comments and Suggestions for Authors

This is a well thought out study. It is an updated account of UBE3A and its detailed mechanistic consideration in ASD. Authors have choreaographical reviewed the topic. I will recommend ''Accept as is''